# Insecticidal Activity of *Angelica archangelica* Essential Oil and Transcriptomic Analysis of *Sitophilus zeamais* in Response to Oil Fumigation

**DOI:** 10.3390/biology14111606

**Published:** 2025-11-17

**Authors:** Guochang Wang, Xing Ge, Dongbiao Lü, Ziyue Zhang, Li Wang, Saili Sun, Xiaoyi Jia, Baizhong Zhang, Kassen Kuanysh, Sarsekova Dani, Hongliang Wang

**Affiliations:** 1Henan Province Engineering Research Center of Biological Pesticide & Fertilizer Development and Synergistic Application, College of Plant Protection and Environment, Henan Institute of Science and Technology, Xinxiang 453003, China; wgchslbh@163.com (G.W.); zzy_94983@163.com (Z.Z.); 15617179499@163.com (L.W.); s17839425509@163.com (S.S.); jxy2025hist@163.com (X.J.); baizhongok@163.com (B.Z.); wanghlzb@163.com (H.W.); 2Hebi Institute of Engineering and Technology, Henan Polytechnic University, Hebi 458030, China; 3Plant Biotechnology Center, Kazakh National Agrarian Research University, Almaty 050040, Kazakhstan; kassenkuanysh12@gmail.com; 4Faculty of Forestry and Land Resources, Kazakh National Agrarian Research University, Almaty 050040, Kazakhstan; dani999@mail.ru

**Keywords:** *Sitophilus zeamais*, essential oil, insecticidal activity, enzymatic activity, RNA-seq

## Abstract

The maize weevil, *Sitophilus zeamais*, is a globally distributed stored-grain pest that causes significant losses to cereals, posing a serious threat to food security. We investigated chemical composition and fumigation activity of *Angelica archangelica* root essential oil (AAEO) against *S*. *zeamais*. The main components of AAEO and its median lethal concentration (LC_50_) for fumigation against this pest were identified. The enzymatic activities of acetylcholinesterase, glutathione-S-transferase, and carboxylesterase were measured when fumigated by AAEO LC_50_. The study demonstrates that AAEO exhibits strong insecticidal activity against *S*. *zeamais*, affecting survival, detoxification enzymes, and gene transcription. The present study provide a scientific basis for the development of effective plant essential oil-based insecticide against *S*. *zeamais*.

## 1. Introduction

*Sitophilus zeamais* is a destructive pest of stored grain, including rice, maize, wheat and their products, causing extensive loss due to reduction in quality and nutritional value [1,2]. Adults of *S*. *zeamais* can directly lay eggs inside the grains to protect the developing larvae, resulting in grain losses and contamination. The larvae infest the grains, causing entrance of various fungi and bacteria, thereby increasing the risk of contamination [3]. To combat postharvest losses, chemical fumigants including methyl bromide and phosphine were frequently and widely used to control *S*. *zeamais* during storage [4,5]. Although they have brought remarkable economic benefits, long-term use of these chemicals has led to the development of pest resistance. Additionally, the chemical fumigants, such as methyl bromide, have been prohibited due to their environmental pollution [6,7]. Thus, it is critical to explore natural products as potential bioinsecticide sources to control stored-product pests.

Plant essential oil (EO), as one class of important volatile secondary metabolites, is extracted from flowers, seeds, leaves, roots and fruits of plants [8]. EOs exhibits not only high bioactivity but also outstanding advantages such as remarkable efficacy, low toxicity, and short persistence in ecosystems [9]. Up to now, many plant essential oils have been reported to have insecticidal activity [10,11,12]; the effects on *S*. *zeamais* could be classified as repellency, fumigation, contact action. Therefore, EOs can be regarded as a foundation of the development of biopesticides [13]. Plant essential oils, as an emerging “green insecticide,” demonstrate significant potential in the field of stored-grain pest control [9]. Research has progressed from simple toxicity assays to in-depth investigations of complex multi-target mechanisms, providing a compelling scientific foundation for their use as alternatives to traditional chemical fumigants [2]. In comparison, their key advantages include high efficacy and broad-spectrum activity, multi-targeted mechanisms of action, environmental friendliness and biodegradability, relative safety for non-target organisms, and a low tendency to induce pest resistance [7]. Comprehensive research positions them as a promising future direction for integrated pest management.

*Angelica archangelica* L. (syn. *A. officinalis* Hoffm.), well known as garden angelica and folk medicine, is a bi-annual or perennial herb with a strong aroma, so it is often used as a food ingredient. The essential oil from roots of *A*. *archangelica* has multiple biological activities, including antioxidant and antibacterial effects, among other pharmacological effects [14,15,16]. However, there is a lack of evidence regarding the insecticidal activity of *A. archangelica* root essential oil against *S*. *zeamais*.

This study investigates the fumigant toxicity of *A. archangelica* root essential oil against *S*. *zeamais* adults and evaluates its impact on the insect’s detoxification and nervous system enzymes. Furthermore, we performed transcriptome sequencing on *S. zeamais* after fumigation with essential oil for 24 h to identify the genes altered by treatment. The results provide a theoretical basis for the management of stored-grain pests using *A*. *archangelica* root essential oil.

## 2. Materials and Methods

### 2.1. Essential Oil

The essential oil from *A*. *archangelica* was obtained from Ji’An HuaTianBao Herbs Biological Products Factory (purity ≥ 95.00%; Ji’an, China). It was extracted from root of *A*. *archangelica* by hydro-distillation.

### 2.2. Gas Chromatography and Mass Spectrometry (GC-MS)

Components of the essential oil of *A*. *archangelica* were identified by a GC-MS 7890B-5977B system (Agilent Technologies, Palo Alto, CA, USA) coated with a DB-5 MS fused silica capillary column (30 m × 0.25 mm × 0.25 μm). n-Hexane was used as the dilution solvent. The injection temperature was set at 250 °C, with 1 μL of a 10% solution injected each time. The temperature was maintained at 50 °C for 2 min, then increased at a rate of 10 °C per minute to 240 °C, and held for 5 min. The carrier gas was helium at flow rate of 1.0 μL/min. Spectra were scanned from 50 to 550 *m*/*z*. The retention indices were determined in relation to a homologous series of n-alkanes (C7–C40) under the same operating conditions. The components were identified by NIST 14.L and confirmed by comparing the Arithmetic Indices and comparison with authentic standards. Relative percentages of the individual components of the essential oil were quantified on the basis of the peak area.

### 2.3. Insect Culture

*S*. *zeamais* were maintained in the insectarium at Henan Institute of Science and Technology for over three years without exposure to insecticides. The insects were reared in a round glass jar with a 10 cm diameter and a height of 15 cm, containing 0.5 kg of sterilized whole wheat. The glass jars were placed in an incubator maintained at 28 ± 1 °C and 75 ± 5% RH, in total darkness.

### 2.4. Fumigant Toxicity

*S*. *zeamais* adults were used to evaluate the fumigant activity of *A*. *archangelica* essential oils (EOs). Solutions of *A*. *archangelica* EOs were prepared at concentrations of 34, 68, 102, 136, and 170 mg/L using hexane. A 100 μL aliquot of each concentration was dropped onto a Whatman filter paper strip (1.5 cm × 5.0 cm) and evaporated for 1 min at room temperature. It was then suspended in a triangular flask (50 mL). Thirty randomly selected *S*. *zeamais* adults were released into a fumigation bottle containing 3 g of wheat, and the cap was screwed tightly shut. The control consisted of hexane solvent. The numbers of dead *S. zeamais* adults were observed and recorded at 24, 48, and 72 h post-treatment. All treatments and controls were conducted independently five times.

### 2.5. Assessment of Enzyme Activity

We determined the activities of acetylcholinesterase (AChE), glutathione S-transferase (GST), and carboxylesterase (CarE). A series of treatments to *S*. *zeamais* were set with dose of 34, 68, 102, 136, and 170 mg/L of *A*. *archangelica* EOs, respectively. Living insects were collected at 24 h after EO treatment, and used for the extraction of total protein. In order to explore the enzyme activity in different stages, another set of test samples were treated with LC_50_ of EO (164.38 mg/L) at 4, 8, 12, 24, and 48 h, respectively. The live insects after treatment were weighed, rinsed, and homogenized, followed by the addition of 900 μL of pre-cooled normal saline. The mixture was centrifuged at 2500 rpm for 10 min at 4 °C. The supernatant was stored at −80 °C. The entire experimental procedure was performed on ice. The concentration of total protein was determined using the total protein quantitative assay (Nanjing Jiancheng Bioengineering Institute, Nanjing, China). The activity of each enzyme, including AChE, GST, and CarE was tested following the instruction of the AChE, GST, and CarE assay kits (Nanjing Jiancheng Bioengineering Institute), respectively. Three replicates were performed for each treatment and each replicate was performed three times.

### 2.6. RNA Extraction and Sequencing

In order to explore the functional genes of *S*. *zeamais* against *A*. *archangelica* EOs, 164.83 mg/L (LC_50_ of 24 h) EOs was used to fumigate the adults for 24 h. The control was hexane solvent. Living insects were collected and frozen separately in liquid nitrogen and stored at −80 °C for RNA extraction.

Total RNA was sequentially extracted using Trizol reagent (Invitrogen, Carlsbad, CA, USA), followed by RNA integrity assessment with the Bioanalyzer 2100 (Agilent Technologies, Santa Clara, CA, USA), and subsequent library construction.

### 2.7. Transcript Assembly and Gene Function Annotation

Raw reads were firstly processed through removing low-quality reads, reads containing adapter, and reads containing N base to obtain clean reads. Transcriptome assembly was accomplished using Trinity with min_kmer_cov set to 2 by default and all other parameters set default [17]. To obtain the comprehensive information of gene functions, assembled unigenes were annotated according to Nr, Nt, Pfam, KOG/COG, Swiss-Prot, KO, and GO database.

### 2.8. Differentially Expressed Genes (DEGS) Analysis

To identify DEGs in *A*. *archangelica* EOs fumigated *S*. *zeamais*, clean reads were analyzed by DESeq2 R package (1.20.0) to calculate the Padj of differential expression [18]. In addition, in order to control error detection rate (FDR), a normalized absolute log2-fold change (fumigated/control) of 2 with adjusted *p*-value less than 0.05 were set as the threshold for significantly differentially expression.

### 2.9. GO and KEGG Pathway Enrichment Analysis

After classifying the annotation information, the DEGs were associated with the Gene Ontology (GO) for enrichment analysis. GO functional enrichment was considered as the differential expression of genes with a significant enrichment threshold of padj less than 0.05. Kyoto Encyclopedia of Genes and Genomes (KEGG) is a comprehensive database integrating genomic, chemical, and system functional information. The threshold for KEGG pathway enrichment was padj less than 0.05. The ClusterProfiler (3.8.1) software was used to conduct GO functional enrichment analysis and KEGG pathway enrichment analysis on the DEGs [19].

### 2.10. Real Time Quantitative Reverse Transcription PCR (qRT-PCR) Analysis

From the DEGs, 14 genes were selected and a heat map was generated using the heatmap package in R software (Version 3.5.0) based on their expression levels in the treatment and control groups, including five P450 genes, three UGT genes, three GST genes, two CarE genes, and one JHEH gene. qRT-PCR was performed on a Real-time Detection System (QuantStuido 5; Waltham, USA) to verify the reliability of transcriptome. qRT-PCR was performed using a 20 μL reaction volume. The program conditions were 95 °C for 30 s, 40 cycles at 95 °C for 10 s, and 60 °C for 30 s. The primers used are listed in Table 1. Glyceraldehyde 3-phosphate dehydrogenase (GAPDH) was used as a reference gene [20]. Three technical repeats were performed for each sample. The relative expression level of genes was performed using 2^−ΔΔCT^.

### 2.11. Statistical Analysis

The percentage of insect mortality was transformed using the arcsine square root transformation. All experimental data were analyzed using SPSS Statistics 22.0 (https://www.ibm.com/products/spss-statistics) with this method. Based on the assumptions of normality and homogeneity of variance, one-way analysis of variance (ANOVA) was subsequently performed, followed by Tukey’s honestly significant difference (HSD) test. Differences were considered statistically significant at *p* < 0.05.

## 3. Results

### 3.1. Chemical Composition of Essential Oil

The chemical composition of the essential oil is summarized in Table 2. A total of 35 compositions accounting for 96.94% of *A. archangelica* essential oil were identified. The major components were δ-3-Carene (24.26%), Limonene (19.81%) and α-Pinene (14.96%), followed by ο-Cymene (9.94%), β-Pinene (6.55%), Linalool (6.49%), Myrcene (5.60%).

### 3.2. Fumigant Toxicity of the Essential Oil

Fumigation assays were conducted to evaluate the fumigant toxicity of essential oils (EOs) against adult *S. zeamais*. The results demonstrated a significant positive correlation between fumigation activity and dosage at 24, 48, and 72 h post-treatment with essential oil (Table 3). At the same dose, corrected mortality also gradually increased with the extension of time, from 24 h to 48 h to 72 h. The largest dose (170 mg/L) of essential oil caused the corrected mortality of 94.48% after 72 h of essential oil treatment (Table 4). The lethal concentration (LC_50_) values were 164.38, 132.62, and 90.35 mg/L air at 24, 48, and 72 h, respectively.

### 3.3. Enzyme Activity

The activity of three detoxification enzymes, AchE, GST and CarE of *S*. *zeamais* was determined. *A*. *archangelica* essential oil fumigation significantly reduced the activity of AchE, and enzyme activity was negatively correlated with the treatment dosage (Figure 1A). At 24 h after fumigation, the activity of AchE was the lowest, after which it showed an increasing trend but remained lower than that of the control group (Figure 1D). A. archangelica essential oil inhibited the AchE activity in adult *S*. *zeamais*. The effects of A. archangelica essential oil on GST and AChE in adult *S*. *zeamais* were generally consistent, both showing an inhibitory effect on enzyme activity. Enzyme activity was negatively correlated with the treatment dosage, and the lowest GST activity was observed at 24 h after fumigation (Figure 1B,E). *A*. *archangelica* essential oil fumigation significantly reduced CarE activity, with treatment dosage and time having only a minor influence (Figure 1C,F).

### 3.4. Transcriptome Analysis

RNA-Seq was carried out to explore the differential genes of *S*. *zeamais* in response to A. archangelica essential oil. As a result, 6.17, 5.99, 6.65 G clean bases for three fumigated groups, and 5.77, 6.75, 5.77 G clean bases for three control groups were produced in this library (Table 5). The percentage of clean reads exceeded 95% of the total reads in each sample. The Q20 and Q30 values of the samples were all above 98% and 95%, respectively, indicating high-quality sequencing data with low error rates and good overall quality. After quality assessment and data filtering, 37,924 unigenes were obtained using Trinity software (Version 2.15.1). Based on the Nr annotation results, we analyzed the species distribution of the annotated sequences. The results revealed that 76.4% of *Sitophilus oryzae* genes matched those in *S*. *zeamais*, indicating a high degree of genomic similarity between the two species. This was followed by matches to *Rhynchophorus ferrugineus* (2.7%), *Anthonomus grandis grandis* (1.65%), and *Acanthoscelides obtectus* (2%) (Figure 2A). Transcriptomic analysis of *S*. *zeamais* treated with *A*. *archangelica* essential oil identified 3718 significantly differentially expressed genes. Expression pattern analysis revealed that 2265 genes were up-regulated, while the remaining 1453 genes were down-regulated (Figure 2B).

### 3.5. GO and KEGG Pathway Enrichment Analysis of DEGs

Functional enrichment results indicated that DEGs in GO terms were mainly related to carbohydrate metabolic process, hydrolase activity, oxidoreductase activity, and catalytic activity. They mainly belong to biological processes and molecular functions, respectively (Figure 3A). The DEGs were significantly involved in KEGG pathways including those related to drug metabolism-cytochrome P450, insect hormone biosynthesis, retinol metabolism, terpenoid backbone biosynthesis, glutathione metabolism, fluid shear stress and atherosclerosis, and platinum drug resistance (Figure 3B).

### 3.6. qRT-PCR Analyses

The expression levels of SzeaCYP6A14, SzeaCYP6M5, SzeaUGT2C1, SzeaUGT2C10, SzeaGST3, and SzeaJHEH1 were significantly higher in the treatment group compared to the control group (*p* < 0.05), whereas SzeaCYP12V2 showed the opposite trend with significantly lower expression in the treatment group (*p* < 0.05) (Figure 4). As expected, the qRT-PCR results for the 14 genes showed similar differential expression patterns between the treatment and control groups as the RNA-seq data, confirming the reliability of our data and analysis.

## 4. Discussion

Several researchers have examined the chemical constituents of the *A*. *archangelica* roots essential oil. Fraternale et al. reported the major components of the essential oil were α-pinene (21.3%), δ-3-carene (16.5%), limonene (16.4%) [16]. Wedge et al. also reported α-pinene (17.5%), δ-3-carene (16.3%), limonene (8.5%) [21]. Factors such as variety, geographic and seasonal variations, as well as extraction techniques, may account for the quantitative discrepancies between the values presented in this study and those reported in earlier research.

*Angelica* essential oils exhibit insecticidal properties against insect pests. The present study is the first to reveal that *A*. *archangelica* essential oil possesses significant fumigant toxicity against *S. zeamais* adults. Accumulating studies have also reported that the essential oils from other (*Angelica*) species, including *A*. *pubescens* [22], *A*. *sinensis* [23], *A*. *dahurica* [24,25], and *A*. *archangelica* [26], showed deterrent, repellent, and acute toxicity. Main compounds such as δ-3-Carene, α-pinene, and limonene may be possible mortality of factors for *A*. *archangelica* essential oil, which are capable of interfering with AChE, CAT and GST activity [27,28,29]. These enzymes are directly related to the insecticidal mechanism.

Li et al. used *Illicium verum* fruit extracts to fumigate *S*. *zeamais*. Under LD_50_ treatment at 12, 24, 36, 48, 60, and 72 h, the extract exhibited an inhibitory effect on GST, with the inhibition strengthening over time [30]. Shang et al. demonstrated that after fumigating *S*. *zeamais* with *Artemisia annua* essential oil, the α-naphthyl acetate esterase (α-NACarE) activity at 2, 4, 8, 12, and 24 h showed a decreasing trend from 2 to 12 h. Although the activity decreased at 2 h and increased again at 4 h, it remained significantly lower than that of the control group [31]. Chaubey et al. reported the effects of α-pinene and β-caryophyllene on acetylcholinesterase (AchE) activity in *S*. *zeamais*, demonstrating that the enzyme activity in the treated groups decreased significantly. This reveals that α-pinene and β-caryophyllene significantly inhibit AchE activity in *S*. *zeamais* [32]. These findings are largely consistent with the experimental results obtained in this study. The inhibition of enzyme activity by essential oils may occur because certain small molecules in the oil are structurally similar to the enzyme’s natural substrate. These molecules bind to the enzyme’s active site but cannot be catalyzed by the enzyme. By occupying the active site, they prevent the genuine substrate from binding to the enzyme, thereby inhibiting its function. The insecticidal mechanism of essential oils may involve a combination of factors, including neurotoxic effects, interference with nerve signal transduction, modulation of GABA receptors, and cellular and physiological toxicity.

Huang et al. evaluated the effects of terpinen-4-ol fumigation on gene expression in *S*. *zeamais* through RNA-seq analysis, revealing 592 DEGs, among which 308 and 284 genes were upregulated and downregulated, respectively. The DEGs were subjected to GO and KEGG enrichment analyses [9]. KEGG enrichment primarily mapped to metabolic pathways, related to the respiration and metabolism of exogenous organisms [33]. Li et al. extracted total RNA from *S. zeamais* and conducted high-throughput sequencing, reporting GO functional classification and KEGG metabolic pathways [34]. These findings are largely consistent with the experimental results obtained in this study.

## 5. Conclusions

This study comprehensively evaluated the control efficacy of *A*. *archangelica* essential oil against *S*. *zeamais* through GC-MS analysis, fumigation toxicity assays, enzyme activity measurements, RNA-seq sequencing, and qRT-PCR. The research revealed 35 chemical components in the essential oil, its fumigation toxicity to S. zeamais, and its effects on the activities of three detoxification enzymes in the insect. A total of 3718 DEGs were obtained, and their expression patterns were validated. These findings will provide data support for developing novel, eco-friendly, plant-based pesticides.

## Figures and Tables

**Figure 1 biology-14-01606-f001:**
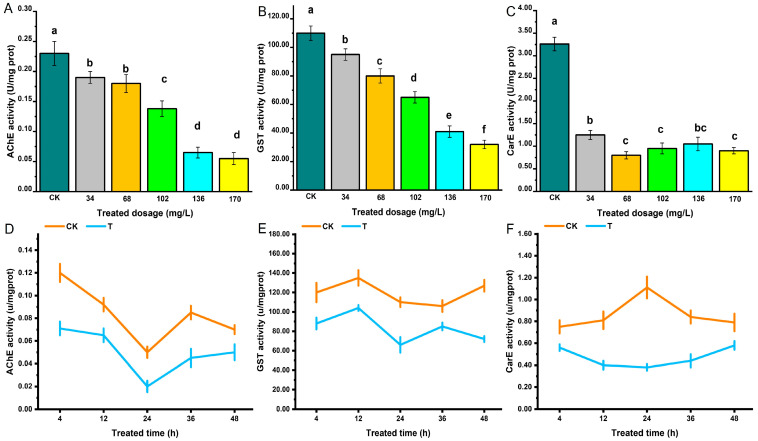
Effects of *A*. *archangelica* essential oil on the Activities of AChE (**A**,**D**), GST (**B**,**E**) and CarE (**C**,**F**) in *S*. *zeamais*. (**A**–**C**) show the effects of essential oil on enzyme activities in 24 h under different dosages; (**D**–**E**) demonstrate the effects of essential oils on enzyme activities at different time points under the LD50 dosage. Different lowercase letters indicate statistically significant differences among treatments (*p* < 0.05).

**Figure 2 biology-14-01606-f002:**
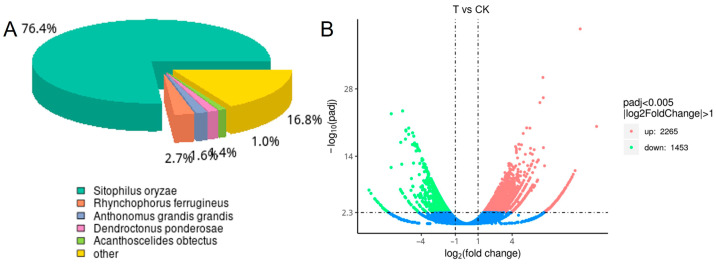
Annotation of species distribution map (**A**) and differential gene volcano map (**B**).

**Figure 3 biology-14-01606-f003:**
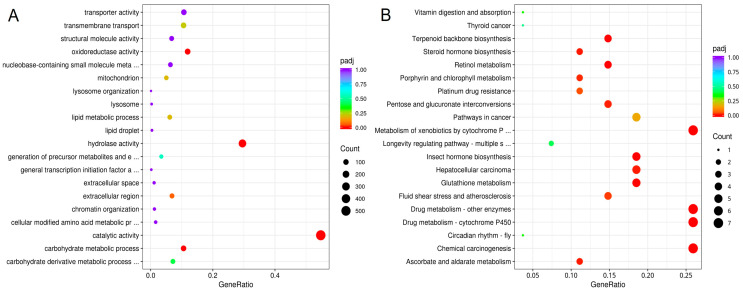
Scatter plot of GO (**A**) and KEGG (**B**) enrichment of DEGs. The horizontal axis represents the ratio of the number of differentially expressed genes annotated to the GO Term/KEGG pathway to the total number of differentially expressed genes, while the vertical axis represents the GO Term and KEGG pathway.

**Figure 4 biology-14-01606-f004:**
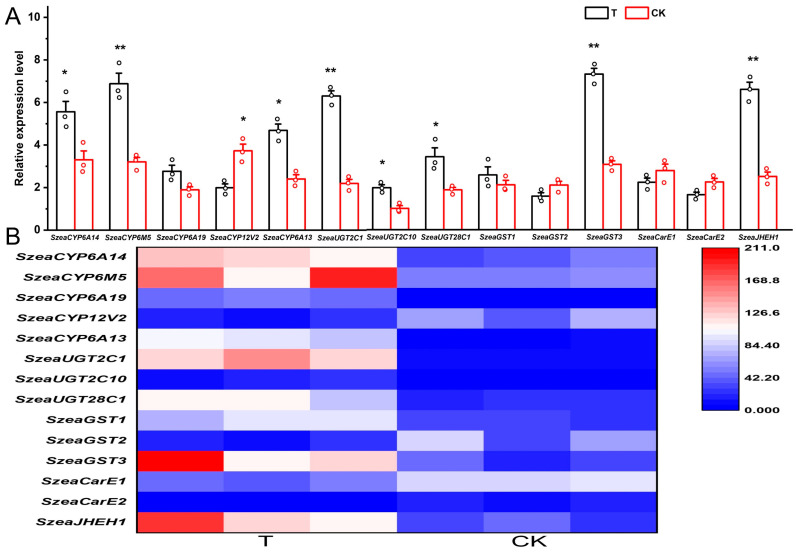
Gene qRT-PCR expression level (**A**) and expression heatmap (**B**). Error bars indicate ± SE. Statistical differences between the treatment and control at the same instar age: *, *p* < 0.05; **, *p* < 0.01.

**Table 1 biology-14-01606-t001:** qRT-PCR primers and internal reference gene sequences.

Primer Name	Primer Sequence (5′-3′)
*SzeaCYP6A14*	F: TCGCTGCTGGATATGAGA
	R: TGGATTGAACTTGTCAGGAT
*SzeaCYP6M5*	F: CTTAGTATCAGCCATCCTAGT
	R: TTGAGTTGCCTGTATTGTTC
*SzeaCYP6A19*	F: GCTAGGCGAGGAATCTTC
	R: CGTAAGCAACAGCAGAATC
*SzeaCYP12V2*	F: GGCTCAGAACAACTCAAGA
	R: GTCCAACCTCCAATTCCAA
*SzeaCYP6A13*	F: ACGAGAAGCGACAGGATT
	R: GAAGAGTGGTTGCGGATG
*SzeaUGT2C1*	F: GTCTGTATCCTGCTCATCC
	R: TGTCCAAGAACTCCTGTAAG
*SzeaUGT2C10*	F: ACCACGACAAGAACTTCA
	R: TGGAACAGTCTGGACCTT
*SzeaUGT28C1*	F: GCACCTTCCATAGTGATGA
	R: CCACGCATTAACAATTCTCT
*SzeaGST1*	F: TGTACGGTATTAAGGCTAGTC
	R: CAGGAATGGTATGTTGAGGA
*SzeaGST2*	F: TAATCTGTCGGAGAAGGAAC
	R: TCGGATAAAGGTCGTCATTT
*SzeaGST3*	F: GCTCTGGAACTTCTCAACA
	R: TTCACCTGCGGATAGTCA
*SzeaCarE1*	F: CGAATCTGTTGAGCAAGG
	R: CAGTTGAGCCGTTGTAAG
*SzeaCarE2*	F: AAGGAGAGTTAGGAGAATCTG
	R: CGTGTTAGTTCATCGTCATC
*SzeaJHEH1*	F: GATGGTGGACTGACAAGA
	R: GACTACAGCCAGAAGGAAT
*GAPDH*	F: AACTTTGCCGACAGCCTTGG
	R: GCGCCCATGTATGTAGTTGG

**Table 2 biology-14-01606-t002:** Chemical composition of *A*. *archangelica* essential oil.

Peak NO.	AI ^a^	Compounds	%RA ^b^	Identification Method ^c^
1	918	Tricyclene	0.15	MS, AI
2	922	Artemisia triene	0.13	MS, AI
3	932	α-Pinene	14.96	MS, AI, AS
4	947	Camphene	1.50	MS, AI
5	967	Sabinene	0.48	MS, AI
6	973	β-Pinene	6.55	MS, AI
7	979	*trans*-Isolimonene	0.23	MS, AI
8	988	Myrcene	5.60	MS, AI, AS
9	998	δ-2-Carene	0.38	MS, AI
10	1010	δ-3-Carene	24.26	MS, AI
11	1013	1,4-Cineole	1.38	MS, AI
12	1023	o-Cymene	9.94	MS, AI
13	1031	Limonene	19.81	MS, AI, AS
14	1068	n-Octanol	0.12	MS, AI
15	1101	Linalool	6.49	MS, AI, AS
16	1127	*cis*-Limonene oxide	0.14	MS, AI
17	1131	*cis*-p-Mentha-2,8-dien-1-ol	0.20	MS, AI
18	1140	Camphor	0.43	MS, AI
19	1165	ρ-Mentha-1,5-dien-8-ol	0.11	MS, AI
20	1181	meta-Cymen-8-ol	0.11	MS, AI
21	1189	α-Terpineol	0.27	MS, AI
22	1197	*trans*-4-Caranone	0.40	MS, AI
23	1213	*trans*-Carveol	0.14	MS, AI
24	1243	Car-3-en-2-one	0.19	MS, AI
25	1282	Isobornyl acetate	0.16	MS, AI
26	1313	*cis*-2,3-Pinanediol	0.48	MS, AI
27	1323	neoiso-Verbanol acetate	0.17	MS, AI
28	1379	*β*-Patchoulene	0.12	MS, AI
29	1386	(2*E*)-Octenol butanoate	0.15	MS, AI
30	1396	Longifolene	0.13	MS, AI
31	1408	(*Z*)-Caryophyllene	0.17	MS, AI
32	1717	(3*E*)-Butylidene phthalide	1.06	MS, AI
33	1944	*m*-Camphorene	0.29	MS, Nist
34	1980	*p*-Camphorene	0.13	MS, Nist
35	2126	(*Z*, *Z*)-9,12-Octadecadienoic acid	0.11	MS, Nist
		Total identified (%)	96.94	

^a^ AI = Arithmetic Index on DB5 column. ^b^ Relative area (peak area relative to the total peak area). ^c^ MS mass spectrum matching with NIST library; AS, authentic sample.

**Table 3 biology-14-01606-t003:** Corrected mortality rate of essential oil of *A. archangelica* against *S. zeamais* adults.

Period (h)	Corrected Mortality Rate (%)				
	34	68	102	136	170
24	6.03 ± 1.06 a	12.19 ± 1.20 b	24.83 ± 1.71 c	38.93 ± 2.46 d	27.72 ± 2.27 e
48	7.53 ± 1.08 a	17.12 ± 1.98 b	32.19 ± 1.67 c	47.26 ± 3.17 b	69.86 ± 3.65 e
72	9.42 ± 1.14 a	23.91 ± 1.62 b	39.96 ± 2.94 c	63.77 ± 4.13 d	93.48 ± 1.35 e

Means in the same row followed by different lowercase letters differed significantly (*p* < 0.05).

**Table 4 biology-14-01606-t004:** Fumigant toxicity of essential oil of *A*. *archangelica* against *S*. *zeamais* adults.

Period (h)	Toxicity Regression Curves	LC_50_ (mg/L air)	95% FL (mg/L)	Relative Coefficient (R^2^)	Chi Square (χ^2^)
24	y = 2.57x − 0.72	164.38	128.02–296.45	0.99	7.90
48	y = 2.85x − 1.07	132.617	104.92–200.42	0.98	9.00
72	y = 3.11x − 1.04	90.35	30.60–226.80	0.99	8.07

**Table 5 biology-14-01606-t005:** Summary of statistical data for the transcriptome of *S*. *zeamais*.

	T1	T2	T3	CK1	CK2	CK3
Raw reads	21,419,493	20,671,023	22,985,184	19,659,294	23,466,508	19,623,581
Raw bases (G)	6.43	6.2	6.9	5.9	7.04	5.89
Clean reads	20,560,357	19,958,351	22,158,451	19,235,246	22,500,960	19,227,394
Clean bases (G)	6.17	5.99	6.65	5.77	6.75	5.77
Clean reads ratio (%)	95.99	96.55	96.40	97.84	95.89	97.98
Q20	98.43	98.55	98.59	98.67	98.61	98.69
GC percentage (%)	36.65	37.54	37.55	43.25	37.33	43.17
Number of unigenes	37,924					

## Data Availability

All the associated data are available in the manuscript.

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
