# Peer review of "Insecticidal Activity of Angelica archangelica Essential Oil and Transcriptomic Analysis of Sitophilus zeamais in Response to Oil Fumigation"

_biology, 2025, doi:10.3390/biology14111606_

Round 1
Reviewer 1 Report
Comments and Suggestions for Authors
Dear Authors,
This study is well designed and thoroughly executed. The conclusions are consistent with established protocols and accurately reflect the study’s objectives and results, as indicated by the title.
The abstract is concise and effectively summarizes the main aspects of the research. The introduction provides a clear and informative overview of the rationale for the study and its potential benefits.
The materials and methods are described in sufficient detail, demonstrating both the methodological rigor and high quality of the work. The results are clearly presented, showing significant potential and opening new directions for future research. However, additional clarification is needed for Figure 1 and Figure 3. The figures are not clearly presented, so that part needs to be corrected.
The discussion appropriately reflects the presented results. The section clearly summarizes previous studies and compares them with current findings. Overall scientific interpretation is correct, but English editing is required for publication quality. The references are adequate in accordance with the topic.
The conclusion is well aligned with the study’s findings and provides a coherent summary of the research.
Overall, this is a well-prepared and valuable manuscript that could be further improved through minor revisions.
Sincerely,
Author Response
Response to Reviewer1 Comments
Comments and Suggestions for Authors:
The materials and methods are described in sufficient detail, demonstrating both the methodological rigor and high quality of the work. The results are clearly presented, showing significant potential and opening new directions for future research. However, additional clarification is needed for Figure 1 and Figure 3. The figures are not clearly presented, so that part needs to be corrected.
Response: Has been modified.
Figure 1 and Figure 3 have been replaced.
Added “Effects of A. archangelica essential oil on the Activities of AChE (A, D), GST (B, E) and CarE (C, F) in S. zeamais. A-C show the effects of essential oil on enzyme activities in 24 h under different dosages; D-E demonstrate the effects of essential oils on enzyme activities at different time points under the LD50 dosage. Different lowercase letters indicate statistically significant differences among treatments (P<0.05).” in line 273 (with tracked changes).
Added “Scatter plot of GO (A) and KEGG (B) enrichment of DEGs. The horizontal axis represents the ratio of the number of differentially expressed genes annotated to the GO Term/KEGG pathway to the total number of differentially expressed genes, while the vertical axis represents the GO Term and KEGG pathway.” in line 310 (with tracked changes).

Reviewer 2 Report
Comments and Suggestions for Authors
Here are some suggestions:
- Please supplement the density of the essential oil and explain how the administration dosage was determined.
- In Section “2.4 Fumigant Toxicity”, no positive control drug was set.
- In Section “3.2 Fumigant Toxicity of the Essential Oil”, the toxicity of oils under different dosages at different time points cannot be intuitively observed from Table 3 provided for the result description. It is recommended to draw scatter plots or other intuitive descriptive forms.
- In Section “3.3 Enzyme Activity”, the annotation of "Figure 1. Effects of Euonymus dahuricus essential oil on the Activities of AChE (A, D), GST (B, E) and CarE (C, F) in S. zeamais" is not clearly described. It is suggested to list them separately after supplementing the complete experimental conditions, such as "Effects of the essential oils on enzyme activities in 24 h under different dosages" and "Effects of the essential oils on enzyme activities at different time points under the LD50 dosage".
- The discussion on the experimental results, especially the insecticidal mechanism of the essential oil, needs to be in-depth, such as analyzing the possible molecular mechanism of the essential oil's inhibition on enzyme activities and the relationship between the enzyme activity and mortality.
Author Response
Response to Reviewer2 Comments
Comments and Suggestions for Authors:
Point 1: Please supplement the density of the essential oil and explain how the administration dosage was determined.
Response 1: Has been modified.
The purity of the essential oil has been confirmed to be ≥95.00%. in line 100 (with tracked changes).
The experimental concentration of the essential oil has been defined.
“Solutions of A. archangelica EOs were prepared at concentrations of 34, 68, 102, 136, and 170 mg/L using hexane.” in line 126 (with tracked changes).
“LC50 of EO (164.38 mg/L) at 4, 8, 12, 24, and 48 h, respectively.” in line 143 (with tracked changes).
Point 2: In Section “2.4 Fumigant Toxicity”, no positive control drug was set.
Response 2: Has been modified.
Thank you for your feedback. In this study, the solvent (n-hexane) was used as the control group, and future studies will increase a positive control using a drug.
Point 3: In Section “3.2 Fumigant Toxicity of the Essential Oil”, the toxicity of oils under different dosages at different time points cannot be intuitively observed from Table 3 provided for the result description. It is recommended to draw scatter plots or other intuitive descriptive forms.
Response3: Has been modified.
Added “Table3. Corrected mortality rate of essential oil of A. archangelica against S. zeamais adults.” in line 254 (with tracked changes).
Table3. Corrected mortality rate of essential oil of A. archangelica against S. zeamais adults
|
Period (h) |
Corrected mortality rate (%) |
||||
|
34 |
68 |
102 |
136 |
170 |
|
|
24 |
6.03±1.06 a |
12.19±1.20 b |
24.83±1.71c |
38.93±2.46 d |
27.72±2.27 e |
|
48 |
7.53±1.08 a |
17.12±1.98 b |
32.19±1.67 c |
47.26±3.17 b |
69.86±3.65 e |
|
72 |
9.42±1.14 a |
23.91±1.62 b |
39.96±2.94 c |
63.77±4.13 d |
93.48±1.35 e |
Means in the same row followed by different lowercase letters differed significantly (P<0.05).
Point 4: In Section “3.3 Enzyme Activity”, the annotation of "Figure 1. Effects of Euonymus dahuricus essential oil on the Activities of AChE (A, D), GST (B, E) and CarE (C, F) in S. zeamais" is not clearly described. It is suggested to list them separately after supplementing the complete experimental conditions, such as "Effects of the essential oils on enzyme activities in 24 h under different dosages" and "Effects of the essential oils on enzyme activities at different time points under the LD50 dosage".
Response 4: Has been modified.
Added “Effects of A. archangelica essential oil on the Activities of AChE (A, D), GST (B, E) and CarE (C, F) in S. zeamais. A-C show the effects of essential oil on enzyme activities in 24 h under different dosages; D-E demonstrate the effects of essential oils on enzyme activities at different time points under the LD50 dosage. Different lowercase letters indicate statistically significant differences among treatments (P<0.05).” in line 273 (with tracked changes).
Point 5: The discussion on the experimental results, especially the insecticidal mechanism of the essential oil, needs to be in-depth, such as analyzing the possible molecular mechanism of the essential oil's inhibition on enzyme activities and the relationship between the enzyme activity and mortality.
Response 5: Has been modified.
Added “The inhibition of enzyme activity by essential oils may occur because certain small molecules in the oil are structurally similar to the enzyme's natural substrate. These molecules bind to the enzyme's active site but cannot be catalyzed by the enzyme. By occupying the active site, they prevent the genuine substrate from binding to the enzyme, thereby inhibiting its function. The insecticidal mechanism of essential oils may involve a combination of factors, including neurotoxic effects, interference with nerve signal transduction, modulation of GABA receptors, and cellular and physiological toxicity.” in line 351 (with tracked changes).

Reviewer 3 Report
Comments and Suggestions for Authors
The paper focuses on using plant essential oils as biological insecticides to control Sitophilus zeamais, which is a research direction with practical application value, especially in the context of the restriction of traditional chemical fumigants such as methyl bromide. The author chooses Angelica archangelica root essential oil to analyze its application in controlling important stored grain pests and the related mechanism of action, which has good theoretical and practical significance. However, there are the following issues in the paper that need improvement:
- The introduction is somewhat vague, and it needs to supplement the related progress in the research of Sitophilus zeamais and the development of pest control methods for stored grain pests, along with an analysis of existing issues. Additionally, there are already many studies on the use of plant essential oils in stored grain pest control, so a stronger summary of this part is needed.
- Enhance the explanation of statistical methods. Although ANOVA is mentioned, the explanation of the specific statistical methods used should be supplemented, such as multiple comparisons, hypothesis testing, etc., to improve the transparency and credibility of data analysis.
- Necessary explanatory notes should be added below figures and tables. For example, in Table 2, what does "AI" mean, and what do the lowercase letters a/b/c represent? Clear definitions and explanations are needed.
Author Response
Response to Reviewer3 Comments
Comments and Suggestions for Authors:
Point 1: The introduction is somewhat vague, and it needs to supplement the related progress in the research of Sitophilus zeamais and the development of pest control methods for stored grain pests, along with an analysis of existing issues. Additionally, there are already many studies on the use of plant essential oils in stored grain pest control, so a stronger summary of this part is needed.
Response 1: Has been modified.
Added “Plant essential oils, as an emerging "green insecticide," demonstrate significant potential in the field of stored-grain pest control [9]. Research has progressed from simple toxicity assays to in-depth investigations of complex multi-target mechanisms, providing a compelling scientific foundation for their use as alternatives to traditional chemical fumigants [2]. In comparison, their key advantages include high efficacy and broad-spectrum activity, multi-targeted mechanisms of action, environmental friendliness and biodegradability, relative safety for non-target organisms, and a low tendency to induce pest resistance [7]. Comprehensive research positions them as a promising future direction for integrated pest management.” in line 74 (with tracked changes).
Point 2: Enhance the explanation of statistical methods. Although ANOVA is mentioned, the explanation of the specific statistical methods used should be supplemented, such as multiple comparisons, hypothesis testing, etc., to improve the transparency and credibility of data analysis.
Response 2: Has been modified.
Added “The percentage of insect mortality was transformed using the arcsine square root transformation. All experimental data were analyzed using SPSS Statistics 22.0 (SPSS, USA) with this method. Based on the assumptions of normality and homogeneity of variance, one-way analysis of variance (ANOVA) was subsequently performed, followed by Tukey's honestly significant difference (HSD) test. Differences were considered statistically significant at P<0.05.” in line 228 (with tracked changes).
Point 3: Necessary explanatory notes should be added below figures and tables. For example, in Table 2, what does "AI" mean, and what do the lowercase letters a/b/c represent? Clear definitions and explanations are needed.
Response 3: Has been modified.
Added “a AI=Arithmetic Index on DB5 column. b Relative area (peak area relative to the total peak area). c MS mass spectrum matching with NIST library; AS, authentic sample.” in line 242 (with tracked changes).
